

# *VCF2PopTree*: a client-side software to construct population phylogeny from genome-wide SNPs

Sankar Subramanian[1], Umayal Ramasamy[1] and David Chen[2]

[1] GeneCology Research Centre, The University of the Sunshine Coast, Sippy Downs, QLD, Australia
[2] School of Information and Communication Technology, Griffith University, Nathan, QLD, Australia

## ABSTRACT

In the past decades a number of software programs have been developed to infer phylogenetic relationships between populations. However, most of these programs typically use alignments of sequences from genes to build phylogeny. Recently, many standalone or web applications have been developed to handle large-scale whole genome data, but they are either computationally intensive, dependent on third party software or required significant time and resource of a web server. In the post-genomic era, researchers are able to obtain bioinformatically processed high-quality publication-ready whole genome data for many individuals in a population from next generation sequencing companies due to the reduction in the cost of sequencing and analysis. Such genotype data is typically presented in the Variant Call Format (VCF) and there is no simple software available that directly uses this data format to construct the phylogeny of populations in a short time. To address this limitation, we have developed a user-friendly software, *VCF2PopTree* that uses genome-wide SNPs to construct and display phylogenetic trees in seconds to minutes. For example, it reads a VCF file containing 4 million SNPs and draws a tree in less than 30 seconds. *VCF2PopTree* accepts genotype data from a local machine, constructs a tree using UPGMA and Neighbour-Joining algorithms and displays it on a web-browser. It also produces pairwise-diversity matrix in MEGA and PHYLIP file formats as well as trees in the *Newick* format which could be directly used by other popular phylogenetic software programs. The software including the source code, a test VCF file and a documentation are available at: https://github.com/sansubs/vcf2pop.

Corresponding author
Sankar Subramanian,
ssankara@usc.edu.au

## INTRODUCTION

One of the major tasks in genetics and evolutionary biology is to infer the ancestral relationship between populations and species. For this purpose, a number of mathematical and statistical algorithms have been developed. To implement these algorithms, computationally efficient software programs were developed. However, these software such as *MEGA* (*Kumar, Stecher & Tamura, 2016*), *PHYLIP* (*Felsenstein, 2005*), *PAUP* (*Wilgenbusch & Swofford, 2003*) and *BEAST* (*Drummond et al., 2012*) are suited only for gene-based sequence data. In the recent past, a series of programs such as *RaxML*

(*Stamatakis, 2006*), ExaML (*Kozlov, Aberer & Stamatakis, 2015*) and MP-EST (*Liu, Yu & Edwards, 2010*) have been developed to infer phylogenetic relationship using whole genome data. A number of sophisticated tree-building software such as *TreeMix* (*Pickrell & Pritchard, 2012*) and QPgraph (*Patterson et al., 2012*) have also been developed to accommodate the number of potential admixture events while inferring the phylogeny. However, to use these software programs, the genome data need to be processed in specific formats such as alignments or allele frequencies.

With the advent of the next generation sequence techniques, large-scale whole genome data containing millions of SNPs are generated for populations. The whole genome data is typically presented in the Variant Call Format (VCF) and there was a need for genetic software to construct population phylogeny that directly use this data format. To address this limitation a number of software programs have been developed in the recent past. However, these programs are either computationally intensive, time consuming, heavily dependent on third party software or require significant time and resource of a web server.

The software programs that reads VCF data are either standalone or web server-based applications. While some of the standalone programs such as *plink* (*Pickrell & Pritchard, 2012*), *ngsDist* (*Vieiran et al., 2016*) and *VCF2Dis* (https://github.com/BGI-shenzhen/VCF2Dis) estimate pairwise distance matrix from VCF files, others such as *FastMe* (*Lefort, Desper & Gascuel, 2015*) and *MEGA* (*Kumar, Stecher & Tamura, 2016*) infer the phylogeny using the matrix. Hence there was a need for software that read VCF files and draw phylogenetic tree directly. To accomplish this a handful of standalone applications such as *SNPhylo* (*Lee et al., 2014*), *VCF-Kit* (*Cook & Andersen, 2017*), and *VCFtoTree* (*Xu et al., 2017*) have been developed. However, these software pipelines need to be installed in a local computer. Furthermore, these programs are dependent on a series of other software such as *bwa* (*Li & Durbin, 2009*), *samtools* (*Li & Durbin, 2009*) and/or *MUSCLE* (*Edgar, 2004*). Therefore, an adequate level of computer expertise is required to implement and run the standalone programs. On the other hand, web server-based programs such as *SNiPlay* (*Dereeper et al., 2015*), and *CSI Phylogeny* (*Kaas et al., 2014*) take significant amount of time to produce a tree using the data from a VCF file. This is partly due to the time taken to upload the large-data set to a server from the user's local machine, which depends on the web traffic and internet speed. Furthermore, both standalone and server-based applications perform a series of data processing steps through software pipelines, which also cause significant time delay.

Due to the reduction in the cost of sequencing and bioinformatic analysis, it is now possible to obtain processed whole genome data for many individuals. Using standard bioinformatic data processing pipelines most of the sequencing service providers deliver high quality publication-ready genotype data for whole genomes in the form of VCF files. Hence population geneticists now need a simple program that reads this data in VCF files and construct a phylogenetic tree in a short time as there is no need of any data processing routines. Therefore, the current study is aimed to the address this important limitation in genomic research. Hence, we developed a JavaScript based client-side software to infer phylogenetic relationship using genome-wide SNP data.

## METHODS

### Implementation

The software, *VCF2PopTree* was written in JavaScript, which runs purely within the user's computer/browser. This program reads VCF files including compressed (gzipped) files. A VCF file contains genotype information in the form of '0' and '1' to denote reference and alternate alleles. *VCF2PopTree* is designed to read and process the input data line-by-line so it is able to handle large data files without running out of memory. The program considers only biallelic SNPs and ignores insertion-deletions (Indels) and SNPs with missing information (./.). Furthermore, based on the user's thresholds (entered in the textboxes) for quality scores and coverage depths, the program filters SNPs. Using the genotype data, two types of measures namely, genetic and drift distances are calculated. To compute pairwise genetic distance between two diploid genomes four pairwise comparisons are performed, and the average is estimated. For instance, the genetic distance for heterozygous SNPs from two genomes (0/1 and 0/1) is 0.5 (2/4). For estimating drift distance, only the dissimilarity of the allele frequencies is considered and for the above-mentioned comparison the drift distance is 0 as the allele frequencies are the same. The distance estimates obtained for each site or SNP are summed to get the total number of differences for the whole genome. The pairwise matrix of these differences are directly used to construct a phylogeny.

To handle missing data there are two options provided. By selecting *Use SNPs present in all genomes* the program will use only the SNPs (passing the threshold score and coverage) that are present in all genomes. In contrast, selection of the alternative option, *Use SNPs for each pair of genomes* will result in including all SNPs that pass through the filters and are present in at least one the pair of genomes. If the total genome length is provided, the program converts the differences to proportions of differences (*p*-distance) and a Jukes-Cantor correction is also implemented. In this analysis numbers of SNPs with missing information (./.), filtered SNPs (based on user's threshold values for quality scores and depth of coverage) and SNPs with more than two alleles are subtracted from the total genome length. The pairwise divergence matrix is then used to infer the phylogenetic relationship using the *UPGMA* (*Sokal & Michener, 1958*) and the classical *Neighbour-Joining* (*Saitou & Nei, 1987*) algorithms and the resulting tree is presented in the popular *Newick* or parenthetical format. The *Newick* formatted phylogeny is used to draw the tree on the browser using the JavaScript package, d3.phylogram.js. Note that the program requires genotype data from at least four genomes in order to build a tree.

### Features

The main web page of *VCF2PopTree* has three major sections (Fig. 1A). First section primarily performs file reading and pairwise divergence calculations. *VCF2PopTree* reads VCF or compressed (gzipped) VCF files. The user has options to filter SNPs based on quality (*Phred*) scores and depth of coverage. The threshold values have to be entered before loading the input file. If the user changes the threshold values, the input file has to be reloaded again. After the input file is chosen a progress bar is displayed to inform whether the file is being read or the pairwise distance is being calculated, which are the major time-consuming steps. Once the above steps are completed the progress bar informs

the user, who can then choose various options listed in the second section of the program to build and display trees and distance matrices on the third section of the program. The phylogeny could be inferred using all genomes or only a set of selected genomes (at least four) by entering the names in the text area, which appears only if the latter option is selected.

As explained in the implementation section, genetic and drift distances could be obtained by choosing the appropriate radio buttons. The pairwise matrices are calculated using the number of differences, *p*- distance or with *Jukes-Cantor* correction. This is achieved by checking the relevant radio buttons and the genome size has to be provided in the textbox to compute *p*- and *JC* distances. The genome size textbox appears only if the options for *p*- or *JC* distances are selected. There are two radio buttons to infer phylogenetic relationship between populations using UPGMA and Neighbour-Joining algorithms and the latter method produces an unrooted tree. Two more radio buttons are provided to draw the phylogenetic tree in a rectangular or circular style. Apart from drawing trees *VCF2PopTree* also produces the tree file in the popular *newick* format by checking the radio button "Newick format" (Fig. 1C). Finally, this program produces pairwise diversity matrix in the popular MEGA (Fig. 1B) and PHYLIP formats and the last two radio buttons should be used for this purpose respectively. Once the file is read and pairwise distances are calculated, the *Draw* is activated.

## RESULTS

### Performance

*VCF2PopTree* is a simple and straight forward program to use, which requires one click to read the VCF file and compute pairwise distances and another to view the phylogeny of a population. *VCF2PopTree* is designed to run on personal computers with moderate specifications. To display a phylogenetic tree, it takes a few seconds to minutes depending on the number of SNVs as well as the number of samples/individuals. For example, it takes only 29 s to display the phylogeny of 10 individuals based on 4 million SNPs from a VCF file using a *Windows* computer with 8 GB RAM and *Intel Core i5* processor. The display time was 3.57 min for a VCF file with 100 genomes and 2 million SNPs. *VCF2PopTree* is compatible with all population browsers including *Chrome*, *Opera*, *Edge and Firefox* and works equally efficient in *Mac*, *Windows* and *Linux* (*Ubuntu*). Furthermore, it displays the tree in a mobile phone (*iPhone* and *Android*) if the input file size is small.

### Comparison with *Galaxy*

To our knowledge Galaxy (https://usegalaxy.org/) is the only available online software that accepts VCF files from large genomes such as vertebrates and constructs population phylogeny. Hence, we compared the performance of our program with that of *Galaxy*. The speed of execution depends on two factors, namely the number of genomes and the number of SNPs (or sites) in the VCF file. Therefore, we first compared the performance of the two programs using the number of SNPs. Figure 2A shows a linear increase in the execution times of both software with the number of SNPs and the correlation is highly significant for both comparisons ($r = 0.98$ and $0.99$ respectively, $P < 10^{-6}$). However,

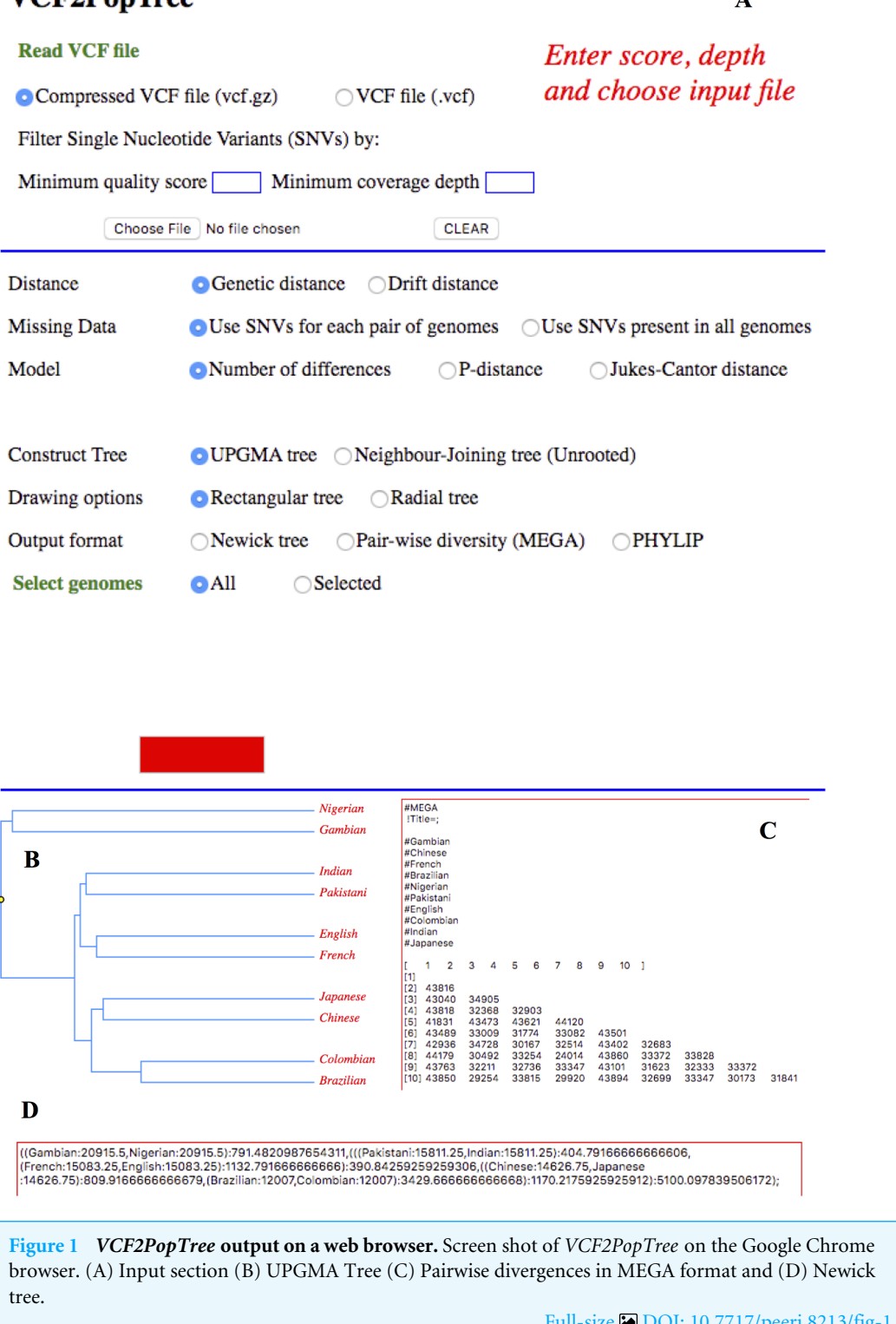

**Figure 1** *VCF2PopTree* **output on a web browser.** Screen shot of *VCF2PopTree* on the Google Chrome browser. (A) Input section (B) UPGMA Tree (C) Pairwise divergences in MEGA format and (D) Newick tree.

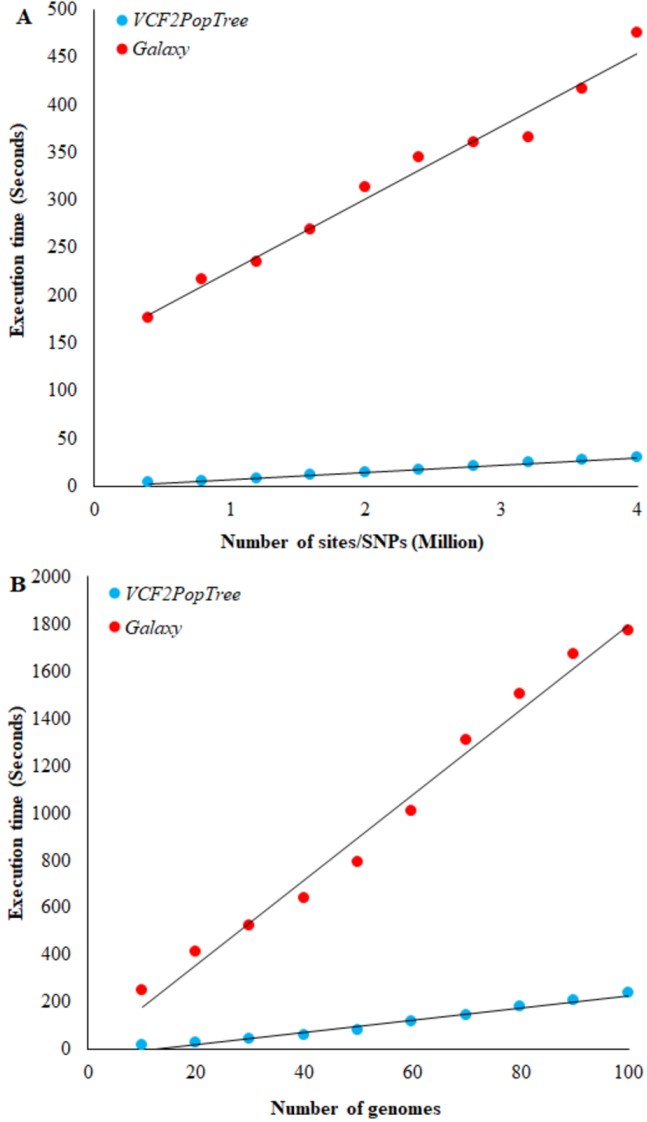

**Figure 2 Comparison of *VCF2PopTree* with the popular online software, Galaxy.** (A) Correlation between the number of SNPs (or sites) and execution times of *VCF2PopTree* and Galaxy (B) Relationship between the number of genomes and execution times. Linear curves best fit the data points. Both correlations are highly significant ($r > 0.98$, $P < 10^{-6}$).

on an average *VCF2PopTree* is an order faster than *Galaxy* in processing the genotype data to produce a phylogenetic tree. While this difference was 58 times for 400,000 lines, it was only 15 times for 4 million lines. Using the equations of the two lines revealed that the difference becomes 10-fold and stays the same (reaches an asymptote) after the number of SNPs reaches 100 million and above. The performance based on the number of genomes revealed a highly significant linear and positive relationship for both software ($r = 0.989$ and $0.992$ respectively, $P < 10^{-6}$) (Fig. 2B). However, *VCF2PopTree* is at least seven-fold faster than *Galaxy* and this difference was 17 times higher for 10 genomes and
7.5 times higher when the number of genomes becomes 100. Extrapolations using the linear equations showed that the difference hovers around 7 times even when the number of genomes is 100,000.

Apart from the slow execution time, to use *Galaxy*, the user has to open a web account and then the VCF file need to be uploaded to the server and converted to *gd_snp* or *gd_genotype* format before obtaining a phylogenetic tree. In contrast, only two clicks are required for *VCF2PopTree* to read and create a tree on a web browser. Therefore, our program is much more user-friendly and immensely useful for users with limited computer skills. Furthermore, the above-mentioned execution time of *Galaxy* was also based on the speed of internet connection as well as the waiting times on the queue. In contrast, internet connection is not required for *VCF2PopTree* as it reads the VCF file from the local machine and there is no issue of queuing in the execution.

## DISCUSSION

Since this is a client-side software, VCF2PopTree.html has to be downloaded to the local computer as "Download Zip" from the GitHub server (https://github.com/sansubs/vcf2pop). To examine the functionality of the software we obtained a compressed VCF file (test.vcf.gz) from the Simons Genome Diversity Project containing about half a million SNPs from ten human populations (*Mallick et al., 2016*). The input file is read by clicking the *choose file* button after providing the values for quality and depth of coverage in the appropriate text boxes. Without those values the program considers all SNPs for further analysis. Once the input file is selected a progress bar appears to indicate the status. After the file is read, pairwise distances are calculated and kept in the memory of the program. The user can then select relevant radio buttons, enter the names of the genomes and genome size and click the button *Draw*. If no names are entered in the text area for genome selection the program will use all genomes. Similarly, if the genome size is not provided phylogeny is inferred based on the number of pairwise differences. The phylogenetic tree or the text area containing the pairwise distance matrix or *newick* tree format is displayed at the third section of the program beneath the *Draw* button. The display can be redrawn multiple time by changing different options without reading the input file as the pairwise distance matrix has already been stored in the memory. A number of alert windows show up if correct VCF files formats are not selected, incorrect names of genomes were entered, or genome size was not entered for calculating *p*- or *JC* distance. The pairwise diversity matrix could be copied and pasted on to a text file, which could be used as an input for programs such as MEGA, PHYLIP or any other software that accepts these formats. Hence users are able to use MEGA and other popular gene specific software to edit or manipulate trees based on whole genome data. Similarly, the whole genome based *newick* tree generated by *VCF2PopTree* could be further manipulated by the tree editing software such as *TreeGraph* (*Stover & Muller, 2010*) or *FigTree* (http://tree.bio.ed.ac.uk/software/figtree/).

# CONCLUSIONS

*VCF2PopTree* is unique with respect to handling whole genome data from populations and it reads data directly from the local machine and is independent of operating systems and browsers. Importantly, this program does not require high performance computational resources, third party software tools, a web server or internet connectivity. It is the fastest software available at present to infer and draw population phylogeny in seconds to minutes. *VCF2PopTree* also produces pairwise distant matrix and *newick* trees, which could be used as the input for the programs such as *MEGA* or *PHYLIP* and thus facilitates whole genome based phylogenetic analysis through other popular software. Therefore, *VCF2PopTree* could be a valuable phylogenetic tree building software for researchers and students in the fields of Genetics, Ecology, Evolutionary Biology and Medicine. *VCF2PopTree* is specifically developed to construct phylogenetic trees for whole genome population data and not for that of species. Therefore, this software is not suited for obtaining phylogenetic tree for species data.

# ACKNOWLEDGEMENTS

The authors are thankful to all who tested the various versions of this software.

### Funding

This project was funded by a Linkage grant awarded to Sankar Subramanian by the Australian Research Council (LP160100594). The funders had no role in study design, data collection and analysis, decision to publish, or preparation of the manuscript.

### Grant Disclosures

The following grant information was disclosed by the authors:
Linkage grant awarded to Sankar Subramanian by the Australian Research Council: LP160100594.

### Competing Interests

The authors declare there are no competing interests.

### Author Contributions

- Sankar Subramanian conceived and designed the experiments, performed the experiments, analyzed the data, contributed reagents/materials/analysis tools, prepared figures and/or tables, authored or reviewed drafts of the paper, approved the final draft.
- Umayal Ramasamy performed the experiments, authored or reviewed drafts of the paper, approved the final draft.
- David Chen analyzed the data, contributed reagents/materials/analysis tools, authored or reviewed drafts of the paper, approved the final draft.

## Data Availability

The source code of the software, test VCF file and the documentation is available at GitHub: https://github.com/sansubs/vcf2pop.

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
