# Peer review of "VCF2PopTree: a client-side software to construct population phylogeny from genome-wide SNPs"

_PeerJ, doi:10.7717/peerj.8213_

## Round 0.1 · original submission · Major Revisions

Please take into consideration all the issues raised by reviewers 1, 3 and 4

·

Basic reporting

The manuscript requires some careful proofreading at the very least. Among other typos:
- people from Colombia are Colombians, not Columbians
- in the abstract, we can read gnome, instead of genome
- the "Simons Genome Project" should read as the "Simons Genome Diversity Project".
- etc

A few references are missing, even when programs are cited, including for bwa, MUSCLE and samtools.

Some sentences are simply not true, such as "these programs are not suited for
large-scale whole genome data" (second sentence in the abstract). It is plenty of tools for inferring NJ trees, most notably including ngsDist or plink (to calculate pairwise distances from NGS data), and fastme to infer a tree from a matrix of pairwise distances.

Some aspects are not totally clear from the manuscript, regarding the calculation of pairwise distances. I expose them in the "general comments to the author"

Experimental design

The main motivation of the paper is to present another user-friendly tool for phylogenetic inference from a VCF file. Although not novel, my opinion is that user-friendly tools are always welcome, especially if they can be easily run as javascript client applications. I would suggest to add some extra functionalities, such as reading compressed VCF files (users do not want to uncompress huge files in their personal computers), calculating genetic and drift distances, the possibility of re-rooting the tree, or to display the branch lengths on top of the corresponding branches.

Validity of the findings

The authors claim that their implementation is faster than others. However, I find some issues a bit unclear:

- the running time depends on two independent variables (i) the number of individuals, and (ii) the number of sites. The computational time is expected to grow quadratically with the number of individuals, because it involves calculating all pairwise distances. I suggest the authors explore their running times as a function of these two variables, separately. For a reader with little expertise in bioinformatic analyses, the file size (in Gb) is a bit cryptic.

- In fig2, the running time seems to grow exponentially for vcf2poptree, while only linearly for the solutions implemented in Galaxy. I suspect then that at some point Galaxy will perform better than vcf2poptree. It is interesting to know when.

Additional comments

Subramanian and colleagues present a user-friendly tool, vcf2poptree, which runs as a client application in personal computers. Although many scientist are nowadays comfortable with existing tools, even if they require some basic bioinformatic skills, I truly appreciate the effort of adapting analytical tools to wider audiences. However, I have serious issues with the clarity of the manuscript, as is in its present form:

1) I think some statements are simply not true, or at least not correctly phrased, such as "these programs are not suited for
large-scale whole genome data" (second sentence in the abstract).
It is plenty of tools for inferring UPGMA and NJ trees, most notably including ngsDist or plink (to calculate pairwise distances from NGS data), and fastme to infer a tree from a matrix of pairwise distances. There are also wonderful and more sophisticated models for calculating trees with potential admixture, such as TreeMix. All these tools scale perfectly well to NGS data. Please, cite all relevant literature.

2) Regarding the calculations of pairwise distances:
- What is the distance between two heterozygous individuals? 0.5 (i.e. two out of the four chromosome comparisons are different)? or 0 (i.e. as the allele frequency is the same in both individuals)? This is very important, because the former case will provide "genetic distances", while the latter "drift distances" (changes in the allele frequency). Conceptually different, both are correct. It would be a good idea to implement both.
- What does happen with non-SNP variants, such as InDels, or with SNPs not passing quality filters? Are they filtered out?
- Does it read polymorphic sites with more than two variants?
- Does the program take into account uncertainties in the genotype calls, if present in the VCF file (e.g. GP field)?
- The NJ method implemented corresponds to the classical algorithm, or includes the improvement presented in Gascuel 1997 MBE?
- Why not implementing at least the Jukes and Cantor correction? It is just adding an equation in the code.
- The authors provide a textbox to specify the total genome size. This is understandable, because the VCF only contains polymorphic positions (usually), and genetic distances are expected to be given by site. However, it is very common to filter out some SNPs during NGS quality processing. The positions corresponding to filtered SNPs should be subtracted from the total genome size. They cannot count neither as monomorphic nor polymorphic, because their status is precisely ambiguous. It is the same for positions that, for whatever reason, are not successfully sequenced (depth = 0). They also need to be subtracted from the total genome size.

Please, clarify all these in your manuscript. As users, we need to know what we are calculating to better interpret the results.

3) Running times (fig 2):
- the running time depends on two independent variables (i) the number of individuals, and (ii) the number of sites. The computational time is expected to grow quadratically with the number of individuals, because it involves calculating all pairwise distances. Although it is implicitly discussed in the main, I suggest the authors regenerate Fig2, evaluating running times as a function of these two variables, separately. For a reader with little expertise in bioinformatic analyses, the file size (in Gb) is not informative, and I would like to see if the tool is appropriated for analysing hundreds or thousands of individuals.

- In fig2, the running time seems to grow exponentially for vcf2poptree, while only linearly for the solutions implemented in Galaxy. I suspect then that at some point Galaxy will perform better than vcf2poptree. It is interesting to know when.

4) I also suggest you carefully proofread your manuscript, as I catched some typos:
- people from Colombia are Colombians, not Columbians
- in the abstract, we can read gnome, instead of genome
- the "Simons Genome Project" should read as the "Simons Genome Diversity Project".
- etc

5) Others:
- A few references are missing, even when programs are cited, including for bwa, MUSCLE and samtools.
- As a potential user, I would like see some extra functionalities, although I understand some require substantial efforts. For example, reading compressed VCF files (users do not want to uncompress huge files in their personal computers), calculating genetic and drift distances, the possibility of re-rooting the tree, or to display the branch lengths on top of the corresponding branches. Also a bar showing the status of the calculations would be nice for large files (progress bar or similar).

Thanks for your effort. I prefer to sign my reviews. My name is Pablo Librado

Reviewer 2 ·

Basic reporting

Excellent manuscript. It is indeed clear with adequate references and excellent results and comparisons.

Experimental design

Well suited for the software article. Direct comparisons are provided.

Validity of the findings

Meaningful comparison provided. All the data are standard and tools are standard.

Additional comments

An important need is addressed. The new software has the potential to become high impact.

Reviewer 3 ·

Basic reporting

The authors provide a JavaScript-based application, VCF2PopTree, to infer phylogenetic trees (UPGMA or Neighbor-Joining) from variation data in VCF format.

Even though the text is well written in general and both the introduction and literature cited are complete and relevant, some suggestions for improvement are:
1. Some sentences should be re-phrased to improve clarity: lines 50-51, 74-75, 86-87, 166-168.
2. The text in the introduction should be written in the present tense in general (e.g. lines 44-47).
3. I think that whenever the authors refer to inferring phylogenies, they should say “infer” and not “deduce” (e.g. lines 12, 36, 166).
4. Line 54: add “is” in “an adequate level of computer expertise is required […]”
5. Lines 140-145: this text is unnecessary
6. Line 165: I would not refer to VCF2PopTree as “ultrafast”.
7. Figure 2 is not cited in the text and I think it is irrelevant, mostly because the running time of Galaxy will depend on the queue and vary from time to time.

Experimental design

The pros of VCF2PopTree are:
1. It is easy to run and does not require any installation.
2. It is independent of the operating system and browser.
3. Maybe useful for non-bioinformaticians as a first glimpse of phylogenetic relationships on their variation data.

However, there are some important cons:
1. It only infers trees by UPGMA and Neighbor-Joining.
2. It does not apply any correction to the genetic distances (e.g. Jukes-Cantor, Kimura). This is basic and very easy to add.
3. There is no information on how missing data is handled.
4. The web-based interface is too simplistic.
5. It does not have a “running” message. Then, and given that results are not displayed instantaneously even for the test data, the user thinks the application is not working, as nothing happens apparently when you upload the input data.
6. It takes time even with the tiny example file. I am not sure how stable the application would be with large datasets.
7. It has few graphical options, basically change from a rectangular to a radial phylogenetic tree, and data needs to be reanalyzed to change from one to the other. I think you should upload input data only once and get both the graphical tree and textual results (Newick and pairwise distance matrices) altogether, and then be able to change graphical aspects instantaneously in one click: e.g. change from rectangular to radial tree, plus other options that VCF2PopTree does not consider now: e.g. include/exclude populations, root on a branch, etc.
8. Instead of entering the genome size, the user may be allowed to choose a genome version (and VCF2PopTree gets the genome size from it).
9. The documentation of the package is very poor.

Finally, the largest problem I see is that in order to perform what VCF2PopTree does, I think it would be more convenient to first calculate a distance matrix from a VCF file (e.g. by using the package VCF2Dis), and then open it in MEGA, where you have many more analysis and graphical options.

Validity of the findings

The MS states that the example dataset contains variation data from ten human populations from the Simons Genome Project. Figure 1 shows the results of analyzing the data with VCF2PopTree.

However, by opening the source file you see that variation data is from chromosome 1 only. This should be stated in the manuscript. In addition, in Figure 1 you can see that the Genome size used as input corresponds to the whole human genome size and not the size of chromosome 1, so the genetic distances displayed in the Figure are meaningless. Finally, the input form of the application is displayed three times, one in each part of the figure, and I think this is unnecessary. At most I would include the form once.

Additional comments

No comment

Reviewer 4 ·

Basic reporting

The manuscript is well written.

Literature is sufficient.

In my opinion, it is more useful to convert the VCF format into sequence or SNP data format in order to do a number of population structure studies using more sophisticated applications. Conversion of VCF to fasta format is already performed by SAMtools.

The Figure 1 is relevant in order to explain the application. The second Figure is also relevant in the context of having a simple and fast way to make a tree.

Experimental design

no comment

Validity of the findings

A validation of the results using alternative applications should be mentioned in a sentence.

Additional comments

The authors develop a software for making NJ/UPGMA trees from VCF files using a simple user-friendly page. There are many applications focused on constructing phylogenetic trees, whose use a wide number of evolutionary models of divergence correction. Nevertheless, those applications are not using the VCF format as input file but they use sequence or SNP data formats. The main reason to not use VCF format is because in phylogenetic reconstruction studies, the sequences come from different experiments, that is, the different species sequences are not sequenced and mapped together. In this work, the authors only focus on individuals of the same species or very close related species (the distances are not considering evolutionary models for correcting divergence) to identify the (population) structure of the samples. The authors provide a simple application to do this analysis using VCF format.

If the application user only wants to do a phylogenetic tree or a pairwise distance matrix, this application is a fast alternative to other applications. Nevertheless, there are a number of issues that this application should consider:

- the application ask for the length of the complete sequence analysed. This information, that is not provided in the VCF format is key in the estimation of the pairwise distances that is provided by the application. I would recommend to avoid using the total length and simply use the pairwise distance of the total region (without dividing by the total positions). The reason is because this information is hard to obtain and the user can be confused by the total length of the genome/chromosome (which may contain large fragments of not sequenced regions or gaps), giving incorrect distances.

- if the authors want to show an analysis about the relationships among individuals, it would be also useful to perform a bootstrap analysis in order to give support to the observed structure. Bootstrap may be useful in case using limited SNPs data.

- The application should read compressed VCF files, at least for gzip format. It is quite painful to unzip a big file in a personal computer. There are available libraries that allow to read these kind of files.

- It is not clear how the missing data is used in this application. I assume that a given SNP position will be used only if all individuals are sequenced at this position.

- The application starts to analyse the data once the VCF file is introduced. It implies that the user has to introduce the information in a sequential way (at least the last must be the VCF file). I would add a new button that indicates the start of the analysis, which would allow to modify the previous parameters/outputs.

In summary, the application is very simple, there are alternatives to do the same analysis, but it is a fast way to do a tree, which may be useful for a number of users not interested in sophisticated structure analysis.

---

## Round 0.2 · Minor Revisions

Please take care of the remaining issues detected by the reviewers

·

Basic reporting

The authors addressed most of my comments, in what I think it is a much better version of the software and the manuscript. I suggest an additional implementation, easy but useful. If the authors agree, the paper can be accepted without sending it me back again.

More specifically, the authors state that sites with missing data are filtered out. I personally think this might be a bad strategy. With the number of genomes increasing, it is possible that a potential user wants to compare hundreds of individuals with their tool. The probability of having at least one individual with missing data growths with the number of individuals compared. So, applying complete deletion, as the authors do, would entail trashing too much data.

An alternative is to add an option for pairwise deletion. This is, in each pairwise comparison, we remove a site only if it is missing in one of the two samples being processed. If the site is successfully genotyped in the two samples, but missing in a third one, the program should keep and analyse that site. The total number of positions analysed will then depend on each specific pairwise comparison.

Same for positions with low coverage/quality. Why trashing a full SNP if only one individual has low coverage? Cannot the program set as missing this poorly genotyped SNP, and proceed with the pairwise deletion approach described above? I truly think that if authors do not implement that, the number of users will be clearly reduced.

English could be polished a lot, but I leave the decision to the editor.

Experimental design

no comment

Validity of the findings

no comment

Additional comments

The authors addressed most of my comments, in what I think it is a much better version of the software and the manuscript. I suggest an additional implementation, easy but useful. If the authors agree, the paper can be accepted without sending it me back again.

More specifically, the authors state that sites with missing data are filtered out. I personally think this might be a bad strategy. With the number of genomes increasing, it is possible that a potential user wants to compare hundreds of individuals with their tool. The probability of having at least one individual with missing data growths with the number of individuals compared. So, applying complete deletion, as the authors do, would entail trashing too much data.

An alternative is to add an option for pairwise deletion. This is, in each pairwise comparison, we remove a site only if it is missing in one of the two samples being processed. If the site is successfully genotyped in the two samples, but missing in a third one, the program should keep and analyse that site. The total number of positions analysed will then depend on each specific pairwise comparison.

Same for positions with low coverage/quality. Why trashing a full SNP if only one individual has low coverage? Cannot the program set as missing this poorly genotyped SNP, and proceed with the pairwise deletion approach described above? I truly think that if authors do not implement that, the number of users will be clearly reduced.

English style could be clearly polished, but I leave the decision to the editor. Thanks to the authors for the effort.

Reviewer 3 ·

Basic reporting

no comment

Experimental design

no comment

Validity of the findings

no comment

Additional comments

I think the authors have made a great effort in introducing most of the suggestions raised by the referees and that the tool is now much more complete, efficient and user-friendly.

I would suggest just making the web form a little bit more intuitive by re-phrasing some fields and adding some help:
- The filters "Quality score" and "Coverage depth" are filters; then, I would replace the text by "Filter chromosome positions by: Minimum Quality Score [__] Minimum Coverage Depth [__]"
- The box to choose selected genomes only applies if the option "Selected" is selected. Then, I would hide the box and display it only when "Selected" is selected. There are easy JavaScript commands to do so.
- Similarly, "Genome size" only applies under certain conditions, so I would do as above. In addition, "Genome size" is vital to get correct results when it is used, to I would add a small note to help the user know which number is expected there according to the VCF provided (whole size of a genome, or a chromosome, ...)
- "Output format" instead of "Data".

Reviewer 4 ·

Basic reporting

The authors have included most of the comments of the reviewers. Although this software application is not making new or additional analysis (which can be performed using other softwares), it has the advantage of constructing a distance matrix (or a phylogenetic tree) very fast and using a single application from a VCF file.

Presumably (I guess depending on the memory of the computer), this program is able to deal with a large number of genomes in short time, which makes this application useful for having a general overview of the structure and relationships of the individuals included. On the other hand, this program has the limitation of dealing with samples very closely related. The reason is because (i) mapping methods used to obtain a VCF file are working with very related samples to the reference, and (ii) the correction methods for estimating divergence are almost inexistent in this software application (only Jukes and Cantor correction method).

Experimental design

The filtered options (Quality score and Coverage depth) are confusing in the interface. They do not indicate if the value included is filtering less than, less and equal than, equal..

Once the VCF file is chosen and read, the program is available to show the plot or the matrix. Nevertheless, a user non familiar with this program may modify the filtering options, without reading again the VCF file, and click for a redraw (thinking that the filtering will be performed). In my opinion, if the user modify the filtering options, the button of "Draw" should be inactive until the VCF file is again read.

The VCF example file is very simple and uniform. I would recommend to include additional VCF files with variability in the quality depths, scores and missing data, in order to see the differences in the trees performed with these different options.

Validity of the findings

no comments

---

## Round 0.3 · accepted · Accept

I'm satisfied that you have addressed the reviewers' comments adequately.